# Functional Niche Partitioning Occurs over Body Size but Not Nutrient Reserves nor Melanism in a Polar Carabid Beetle along an Altitudinal Gradient

**DOI:** 10.3390/insects14020123

**Published:** 2023-01-25

**Authors:** Diane Espel, Camille Coux, Luis R. Pertierra, Pauline Eymar-Dauphin, Jonas J. Lembrechts, David Renault

**Affiliations:** 1CNRS, ECOBIO (Ecosystèmes, Biodiversité, Evolution), UMR 6553, University of Rennes, F-35000 Rennes, France; 2CNRS, Inserm, CHU Lille, Institut Pasteur Lille, U1019-UMR 9017-CIIL-Center for Infection and Immunity of Lille, University of Lille, F-59000 Lille, France; 3CEFE, University of Montpellier, CNRS, EPHE, IRD, F-34000 Montpellier, France; 4Department of Plant and Soil Sciences, University of Pretoria, Pretoria 0002, South Africa; 5CNRS, LEHNA (Laboratoire d’Ecologie des Hydrosystèmes Naturels et Anthropisés), UMR 5023, University of Lyon 1, F-69100 Villeurbanne, France; 6Research Group Plants and Ecosystems (PLECO), University of Antwerp, 2610 Antwerpen, Belgium

**Keywords:** *Amblystogenium pacificum*, Carabidae, functional diversity, French sub-Antarctic islands, thermal tolerance, melanism, dimorphism

## Abstract

**Simple Summary:**

When exposed to gradual stresses such as climate warming, species with high plasticity may develop different growth types (morphotypes) adapted to specific ranges of the temperature spectrum, thereby securing their survival in various scenarios. *Amblystogenium pacificum* is a carabid beetle endemic to the sub-Antarctic Crozet Islands that has two distinctive morphotypes based on body coloration. Here, we measured functional traits related to morphology and biochemical reserves to test whether they were related to morphotype, sex, and altitude (as a proxy for a temperature gradient). We also tested whether distinct functional niches (trait profiles) could be identified based on morphotype, sex, and altitude. We found a positive correlation between altitude and body size as well as higher protein and sugar reserves in females than in males. The body size profile showed the clearest functional response of *A. pacificum* along the altitudinal gradient, even though darker morphotypes tended to be smaller and more constrained at higher altitudes and females showed limited trait variations at the highest altitude. Hence, our results show mitigated responses of *A. pacificum* in the temperature–size relationships such that further trait measurements are necessary before using this as a model case to investigate population range shifts in relation to global changes.

**Abstract:**

Phenotypic plasticity can favor the emergence of different morphotypes specialized in specific ranges of environmental conditions. The existence of intraspecific partitioning confers resilience at the species scale and can ultimately determine species survival in a context of global changes. *Amblystogenium pacificum* is a carabid beetle endemic to the sub-Antarctic Crozet Islands, and it has two distinctive morphotypes based on body coloration. For this study, *A. pacificum* specimens of functional niches were sampled along an altitudinal gradient (as a proxy for temperature), and some morphological and biochemical traits were measured. We used an FAMD multivariate analysis and linear mixed-effects models to test whether these traits were related to morphotype, altitude, and sexual dimorphism. We then calculated and compared the functional niches at different altitudes and tested for niche partitioning through a hypervolume approach. We found a positive hump-shaped correlation between altitude and body size as well as higher protein and sugar reserves in females than in males. Our functional hypervolume results suggest that the main driver of niche partitioning along the altitudinal gradient is body size rather than morphotype or sex, even though darker morphotypes tended to be more functionally constrained at higher altitudes and females showed limited trait variations at the highest altitude.

## 1. Introduction

Abiotic factors, and in particular climatic characteristics, are widely acknowledged as some of the critical drivers of the distribution of ectothermic species [1,2,3]. As most of these environmental factors vary across habitats and seasons, the plasticity of life history traits, or phenotypic plasticity, is essential to allow timely adjustments that maintain organisms at an optimal fitness and eventually determines whether species survive in heterogenous and variable environments [4,5]. By allowing organisms to adjust to various environmental conditions, phenotypic plasticity expands the range of potential future habitats and prepares species for temporal changes, i.e., two processes particularly important under global change perspectives [4,6]. Interestingly, when the variation in environmental factors exceeds species’ physiological thresholds, intraspecific plastic responses can contribute to the emergence of different ecotypes, i.e., distinct populations that are adapted to specific environmental conditions, which are often driven by singular morphological or physiological traits [7]. Each ecotype is thus characterized by a greater ability to develop and exploit a specific range of environmental conditions, and as such, outcompetes relatives from distinct ecotypes. 

Often, traits are considered to highlight the functional responses of organisms to the abiotic characteristics of their habitats [8,9]. In this context, functional diversity, which is a measure of the size of the functional niche and is thus directly correlated to phenotypic plasticity, has increasingly been studied [10,11,12]. At the species or community scales, high functional diversity is often associated with higher resilience to environmental changes [8,13,14,15]. 

Size-related morphological traits are ultimately correlated to body size, which in turn is intimately linked to an organism’s fitness. Hence, the study of body size variations along ecological clines has received much attention over the years [16], particularly along temperature gradients [17,18]. For endothermic species, larger body sizes often correspond to a smaller surface area to volume ratio and reduced heat dispersion in colder climates, a pattern defined as Bergmann’s rule [19]. For ectotherms, latitudinal and altitudinal gradients have historically been considered proxies for temperature characteristics, such that larger ectotherms may be observed at higher latitudes and altitudes [20]. Although a broad 80% of ectotherms are estimated to follow Bergmann’s rule, some notable exceptions occur, especially among insects. Indeed, there is a growing body of evidence that reports unclear patterns [21,22] or no patterns at all [23,24] in body size changes along altitudinal gradients. Counterexamples even showed that insects exposed to cold [25] or thriving at high altitudes [26,27] had reduced body sizes.

Additionally, sexual size dimorphism (SSD) is a common feature in many animals, including insects [28]. Female insects are usually larger than males, but in a systematic review, Teder and Tammaru [29] found considerable variation in the degree and even the direction of intraspecific SSD, which they linked to environmental conditions. Rensch’s rule, which is not limited to insects, posits that SSD increases with body size when males are the larger sex but decreases when females are larger [30,31]. Moreover, body size is often correlated to the standard metabolic rates (SMRs) of insects, where larger insects should have lower SMRs and higher amounts of lipids [32]. Consistently, the meta-analysis conducted by Lease and Wolf [33] reported that lipid contents scaled isometrically with body mass protein contents and decreased with elevation but only in females. SSD could therefore be another confounding factor influencing the relationships between temperature or other environmental gradients and body size. 

Body coloration can influence different functions in insects, including thermoregulation. Higher melanism is a common response among insects that are exposed to cold conditions, and darker insects are often prevalent when their habitats include cold environments [27,34,35,36]. This pattern is known as Gloger’s rule, which links animal body coloration to climatic conditions (see [37] for a review). The thermal melanism hypothesis posits that insects with darker body colorations and reduced body surface reflectance values are favored at low temperatures. This may indeed help them to warm up more efficiently [35] and increase locomotion speed [38]. 

The carabid beetle *Amblystogenium pacificum* Putzeys 1869 (Carabidae), which is endemic to the Crozet Archipelago [39,40,41], is an insect species that has two distinct morphotypes. Some populations of the beetle are dominated by brownish-colored adults (‘light morphotype’), while others are predominantly black (‘dark morphotype’). According to Davies et al. [42], the proportion of dark individuals increases by about 8% every 100 m in altitude on Possession Island (Crozet Archipelago), resulting in a clear distinction in the spatial distributions of the two morphotypes. Recent observations made in fellfield habitats reported that the brown to black proportion ranged from a ⅔ ratio of brown insects at low altitudes to a ⅔ ratio of black adults at higher altitudes (IPEV136 Project SUBANTECO, Renault D., unpublished observations), which suggests at least partial ecotypic segregation and a potential shift in the performances of functional traits along the altitudinal gradient. 

Some studies have investigated the ecology and biology of *A. pacificum* [40,41,42,43], but none have examined if their altitudinal distribution has resulted in body size and coloration differences between males and females. Here, we aimed to determine if there were differences in morphology and body reserves (lipids, proteins, and sugars) among the two morphotypes of *A. pacificum* (trait space segregation by color) in order to determine whether this species could be taken as a model case to evaluate intraspecific resilience to environmental changes. Specifically, following Bergmann’s rule, we expected to find the smaller-sized individuals, with lower amounts of body reserves (lipids, proteins, and sugars), at lower altitudes. As a darker coloration is assumed to confer advantageous thermal tolerance benefits at higher altitudes, we hypothesized that darker adults of *A. pacificum* would also be characterized by higher amounts of body reserves compared to their lighter-colored counterparts. 

To test these hypotheses, we first simultaneously evaluated the effects of altitude, morphotype, and sex on the morphological and biochemical traits through a factor analysis of mixed data (FAMD) analysis. Then, we ran a series of linear mixed-effects models (LMEs), using each trait as a response variable to isolate their raw responses to altitude, morphotype, and sex. Finally, we explored the intraspecific variation in the functional niche of *A. pacificum* based on the projection of measured morphological and biochemical traits into a hypervolume space [44,45,46]. This allowed us to compare the sizes and overlaps of the hypervolumes and determine whether functional niche segregation was occurring between the light and dark morphotypes or males and females, depending on the altitude level. 

These hypervolumes were used to explore (i) if *A. pacificum* followed Gloger’s rule, with darker morphotypes not only being more common at higher sites but also showing a specialized use of the functional niche (as would be revealed by a reduced hypervolume size); (ii) if the species showed sexual dimorphism; and (iii) if this sexual dimorphism varied along the altitudinal gradient. Given that higher proportions of dark morphotypes were recorded at higher altitudes [40], we hypothesized that there could be some niche partitioning among light and dark morphotypes of *A. pacificum* along the altitudinal gradient. Moreover, we expected this partitioning to have an impact on other features of *A. pacificum*, namely the compositions of nutrient reserves and morphological traits. Since these traits are also known to differ between males and females, we also expected niche partitioning between sexes, possibly in an interaction with the altitudinal gradient. Therefore, if the functional niche size decreased at higher sites, an explanation could be that the constraints of the harsher environment selected for the trait values that were most adapted to these conditions. To gain further insight, the exploration of whether the niche space was shared or segregated among morphotypes, sexes, or altitude could be pursued by looking at the degree of niche overlap and how each trait contributed to the functional diversity in each case. 

## 2. Material and Methods

### 2.1. Study Area

This study was carried out on insects collected on Possession Island (147 km², Figure 1), the largest island in the Crozet archipelago (45°48′–46°26′ S, 50°14′–52°15′ E), which is part of the nature reserve of French southern territories and has been listed as a UNESCO World Heritage site since 2019. Possession Island is located in the southern Indian Ocean, 2400 km north of the Antarctic continent and 2400 km south-east of the South African coast, giving it a typical sub-Antarctic climate, with frequent rain and moderately low temperatures (average annual temperature of 5.6 °C) (Météo France 1960–2019 records). Along its mountainous topography (ranging from 0 to 934 m above sea level (asl)), the temperature is estimated to decrease by about 0.7–0.8 °C for every 100 m of elevation [47].

### 2.2. Field Sampling

*Amblystogenium pacificum* Putzeys 1869 usually occupies a mesic fellfield habitat (i.e., rocky and stony mineral soils with relatively sparse vegetation) [40] and can occur along the whole altitudinal gradient of the island. Field sampling was conducted during the southern-hemispheric summer, between December 2017 and February 2018. Adult specimens of both morphotypes were easily observed and manually captured in the inter- and sub-stone spaces on the Crozet Islands. For this study, insects were hand-collected at five sites along an elevation gradient covering the full altitudinal range of Possession Island, i.e., from 0 to 900 m asl (Figure 1), following quotas imposed for the collection of endemic species from the sub-Antarctic Crozet Islands (Arrêté A-2017-113, 12 October 2017, Terres Australes et Antarctiques Françaises).

Each sampling started with a visual estimation of the relative abundance of ‘dark’ (D-morphotype, i.e., black coloration) and ‘light’ insects (L-morphotype, i.e., brown coloration) at a given altitude in order to collect each morphotype in proportions reflecting their relative abundances at each altitude: 12 insects of the ‘dark’ morphotype and 11 insects of the ‘light’ morphotype at 0 m asl, 12 ‘dark’ and 12 ‘light’ insects at both 130 m and 300 m, 26 ‘dark’ and 20 ‘light’ insects at 600 m, and finally 13 ‘dark’ and 9 ‘light’ insects at 900 m asl. All collected specimens were preserved in 70% ethanol at −25 °C until they were processed for the analyses described below. 

### 2.3. Functional Trait Measurements

Functional traits refer to morphological, physiological, and phenotypic characteristics that directly or indirectly influence the performance of any individual in terms of growth, reproduction, and survival (e.g., [9,10,48]). To examine if there were intraspecific variations in the trait values of adult *A. pacificum* (i.e., between the two morphotypes and the two sexes) sampled along the studied altitudinal gradient, several morphological and physiological traits were measured for each collected individual. First, six size features were measured: the interocular distance, as a proxy of head size (Figure 2A); the length and width of the pronotum (Figure 2B); the length of the femur of the third left leg (Figure 2C); and length and width of the elytra (Figure 2D). These measurements were made with the ©ZEN software using photos taken with a camera (AxioCam ERc 5s, ZEISS, Munich, Germany) connected to a binocular magnifier. 

For each individual, we then calculated a body size index corresponding to the coordinates of the individuals on the main axis from a principal component analysis (88.2% of the explained variance). The body size index was computed from the measurements performed on the above-mentioned morphometric traits (cf., Appendix A). The body water contents and ethanol residues of the insects were discarded by drying the individuals (Speed Vac Concentrator, MiVac, Genevac Ltd., Ipswich, UK) for 24 h at 32 °C. Each insect was then weighed (XP2U Mettler Toledo Balance, Mettler Toledo®, Columbus, OH, USA; d = 0.1 μg), and its sex was identified [49]. 

In a second part of the study, biochemical assays were performed to extract and quantify the protein, sugar, and lipid body contents. Each insect was transferred to a 2 mL microtube with 180 μL of phosphate buffer and was ground for 90 s at 25 Hz with a bead beater (RetschTM MM301, Retsch GbmH, Haan, Germany). After centrifugation (10 min, 4000× *g*, 4 °C), a 5 μL aliquot of the supernatant was collected from each sample and transferred to the well of a microplate. A 235 μL volume of Bradford’s reagent was added to each well [50]. The optical densities of the samples were read at 595 nm, and protein quantities were calculated from the calibration curve made from different concentrations of bovine serum albumin diluted in phosphate buffer. The remaining 175 μL of each sample was further mixed with 425 μL of ultrapure water and 900 μL of methanol/chloroform (2:1 ratio, volume/volume). After centrifugation (10 min, 4000 g, 4 °C), a 200 μL volume of the upper phase (methanol and water containing the sugars) was collected and transferred to a new 2 mL microtube. Each sample was vacuum-dried, as described above. The amounts of total sugars were then measured by adding 800 μL of 70% sulfuric acid containing anthrone at 1.42 g·L^−1^. Following incubation at 90 °C for 10 min, a 240 μL aliquot was used to determine the amounts of total sugars with colorimetry (VersaMaxTM Microplate Reader, Molecular devices, San Jose, USA) at 625 nm. A standard curve was also calculated at 625 nm. Finally, the lower phase of the samples, containing the chloroform in which the lipids had been dissolved, was analyzed. A 200 μL volume of the chloroform extract was collected and vacuum-dried (speed-vac) before being redissolved in 200 μL of 0.2% Triton X-100 and 3% bovine serum albumin. A 20 μL volume of this mixture was mixed with 230 μL of a triglyceride reagent to determine the amounts of lipids using colorimetry at 510 nm. A standard curve was also performed at 510 nm. 

For all biochemical measurements, quantities were related to the dry body mass of each individual, and the results are hereafter referred to as proteins/mass, sugars/mass, and lipids/mass. 

### 2.4. Statistical Analysis

All statistical analyses were performed in the R software environment (R version 4.0.5, R Core Team, 2021).

As an exploratory analysis, we first ran a multivariate factor analysis of mixed data (FAMD) [51] using the R packages ‘FactoMineR’ [52] and ‘FactoExtra’ [53] to simultaneously explore the relationships among the functional traits (i.e., the body size index, the body mass, and the amounts of proteins, sugars, and lipids), morphotype, sex, and altitude.

#### 2.4.1. Univariate Approach: Influence of Altitude, Sex, and Morphotype on Functional Traits

We built linear mixed-effect models (‘lme4’ R package [54]), using each of the five continuous traits (body size index, body mass, lipids/mass, sugars/mass, and proteins/mass) as response variables in five separate models. We log-transformed the sugars/mass and lipids/mass traits to approximate a normal distribution. In each of these five models, we tested for the main effects of altitude, sex, and morphotype as well as their two- and three-way interactions. We also included a quadratic term for altitude to test for potential non-linear effects and its two-way interactions with the other variables and added “Site” as a random variable to control for non-independence among insects collected from the same altitude. Then, for each of the five models, we built every possible subset of the full models by sequentially removing covariables and selected models for which Akaike’s information criterion scores were the lowest and did not differ from one another by more than two units, which suggested similar model performance [55]. Finally, we applied model averaging [56] on the selected best-fitting full average models using the ‘MuMIn’ R package [57]. 

#### 2.4.2. Niche Space of *Amblystogenium pacificum*: Functional Hypervolumes

One way of evaluating the functional response of a community or, in our case, of a group of individuals from the same species, to environmental characteristics is to use measures of functional traits to project species (or individuals) into a multidimensional space where each trait corresponds to an axis. The projected entities are functionally similar when located close to one another in the functional space, and the description of the globally occupied functional hypervolume provides information about the functional niche space of the community/population. Importantly, this approach allows the detection of synergies and antagonisms among the measured traits and reveals the relative responses of these traits to environmental characteristics according to one another.

To compare the functional responses of *A. pacificum* to altitude, morphotype, and sex, we calculated the functional niche spaces for corresponding individuals. We used the *n*-dimensional hypervolume approach [58] further described below, with the five aforementioned continuous functional traits as dimensions.

We used the body size index as a proxy for morphological traits, and we scaled and centered the remaining biochemical trait values prior to the computation of the hypervolumes to enable the comparison of trait axes with different units. Then, hypervolumes were computed using the Gaussian kernel density estimation method [44]. This algorithm approximates trait values as clouds of stochastic points sampled from the set of observed trait values. The resulting unit of hypervolume size was the standard deviation of the transformed trait values (i.e., scaled, centered, and for the sugars/mass and lipids/mass traits, also log-transformed to conform to normality requirements) raised to the power of the number of trait dimensions (SD^n^, with *n* axes, i.e., traits). 

To characterize the functional niche space of a given community (in our case, a subset of individuals from the same species), we used three metrics: 

(i) The hypervolume size, which measures the variation in the range of values taken by individuals for each dimension (or trait) and refers to functional richness (alpha functional diversity according to Mammola and Cardoso [12]). A reduced hypervolume size suggests that trait values are more constrained. We compared hypervolume sizes among groups (e.g., between altitude levels) using a *Z*-test. 

(ii) The hypervolume overlap (H_overlap_), which measures the degree to which two niches share the same functional space (i.e., functional similarity or beta diversity). We used the Sørensen–Dice dissimilarity index to compare hypervolume overlaps between altitude levels. 

(iii) The contribution of each trait to the size of the hypervolume, measured as the ratio between the size of the hypervolume containing all traits and the hypervolume without the trait of interest. If the contribution of a given trait in a site is very low, it means that the values from that trait are constrained to a reduced range. One of the reasons for trait constriction could be that the selection for these values is strong, which could suggest niche segregation. 

In order to test these hypotheses, we compared the hypervolume metrics among the five sites located along the studied altitudinal gradient. This was first performed for all morphotypes and sexes pooled together and then between light and dark morphotypes on one hand and between males and females on the other hand. To check for the simple effects of morphotype and sex without the altitude effect, we also compared hypervolumes by pooling sites for each subset.

As a hypervolume size depends on the number of species (or individuals) that were used to calculate it, we bootstrapped the trait values from the subset of individuals of interest and calculated the aforementioned metrics based on 100 resampled hypervolumes [59,60]. This enabled us to calculate confidence intervals and identify the cases where the observed values fell outside of these intervals. As it is difficult to obtain a balanced dataset of individuals, we set the minimal number of individuals needed for simulations to the minimum number of collected specimens per group (e.g., n_minimum_ = 21 random insects per altitude level for pooled data).

## 3. Results

The results of the FAMD analysis are presented in Appendix A. 

### 3.1. Univariate Approach: Influence of Altitude, Sex, and Morphotype on Functional Traits

Altitude had a significant effect on the body size index (Table 1A): the smallest individuals were generally found at the lowest altitudes, and body size peaked at 600 m before declining again at 900 m asl. 

The significant interaction between altitude and morphotype (Table 1A) indicated that this pattern was stronger for dark morphotypes than for light ones (Figure 3A). The three-way interaction between altitude, morphotype, and sex was not significant and was not retained in the model selection process.

The protein and sugar amounts were both higher in females compared to males (Table 1C,D and Figure 3C,D). The protein contents were also higher, on average, for light morphotypes (Table 1C and Figure 3C). Body mass and lipid amounts were not affected by altitude, morphotype, or sex (Table 1B,E and Figure 3B,E), even though a trend was observed: there was a proportional change in body mass in relation to body size and a tendency for sugar amounts to evolve with a negative correlation to body size along the altitude gradient. 

### 3.2. Niche Space at the Species Scale

#### 3.2.1. Functional Diversity of *Amblystogenium pacificum* along the Altitudinal Gradient

The hypervolumes for all insects over the five altitudinal sites were large, relatively diffuse, and occupied similar spaces in the functional niche (Figure 4A; left panel). Niche overlap was relatively important among insects from the lowest altitudes (i.e., between sites located at 11 and 130 m asl, H_overlap_ = 0.419) and among the sites located at altitudes higher than 130 m asl (H_overlap_ from 0.380 to 0.545) (Appendix A). However, niche overlap was lower between low (11–130 m asl) and high (>130 m asl) altitudes (e.g., H_overlap_ = 0.002 between 11 m and 600 m hypervolumes).

The volumes of the insects collected from the 11 m and 300 m altitudes were significantly different from one another, with the latter being the most constrained (Figure 4A; middle panel) (*Z*-test, Bonferroni-adjusted *p*-value = 0.015). The decrease in functional richness at 300 m asl was mainly shaped by the low variation in protein amounts, while high variation in lipid amounts explained the higher functional richness at 11 m asl (Figure 4A; right panel). The biochemical traits generally showed the highest intraspecific variations, in particular the protein and lipid amounts at 600 m and 11 m asl (Figure 4A; right panel). The body size index, body mass, and protein amounts were all beneath the confidence interval limit at at least one site (and at all sites for the body size index). This indicated that the overall variation in these trait values was relatively low within each site.

#### 3.2.2. Morphotype and Sex Functional Trait Space

When pooling all sites together, the hypervolumes calculated for the light and dark morphotypes and for males and females occupied a similar niche space (Figure 4B,C; left panels): the niche overlap was important between the two morphotypes (H_overlap_ = 0.649) and sexes (H_overlap_ = 0.656). 

Nonetheless, the hypervolume of the light morphotypes was significantly larger than for the dark ones (Z-test, Bonferroni-adjusted *p*-value < 0.001) and slightly larger for males than for females (Z-test, Bonferroni-adjusted *p*-value = 0.014). These differences in hypervolume size were not explained by the higher contribution of one particular trait (Figure 4B,C; right panels). Hence, the two morphotypes and sexes occupied similar parts of the functional space but in different ways.

### 3.3. Intraspecific Niche Comparison

#### 3.3.1. Comparison of Morphotypes According to Altitude

Calculating the functional hypervolumes for the light and dark morphotypes separately for each altitude level unveiled differences in the way they occupied the functional space along the altitudinal gradient (Figure 5A,B; left panels). In terms of hypervolume sizes, however, the only significant difference was found between dark insects from the 300 m and 600 m sites (Z-test, Bonferroni-adjusted *p*-value = 0.002). For both morphotypes, the body size index contributed less than the confidence interval to the sizes of the hypervolumes of almost every site, which again showed a lack of intrasite variation in body size (Figure 5A,B; right panels).

#### 3.3.2. Comparison of Sexes According to Altitude

We observed some differences in the functional hypervolumes relative to males and females across the altitudinal gradient (Figure 5C,D; middle panels). Indeed, hypervolume size for females tended to decrease with altitude, with the functional richness of the 130 m site being significantly higher than the functional richness of the 900 m site (Z-test, adjusted *p*-value = 0.011). Conversely, males tended to have a positive quadratic response to increasing altitude, with the minimal volume corresponding to the 300 m site being significantly lower than the hypervolume size of the 11 m site, which was its maximal size, and lower than that of the 900 m site.

As for the contributions of individual traits to the hypervolumes, the only trait to significantly negatively impact the hypervolume size was, again, the body size index, especially for males (Figure 5C,D; right panels).

## 4. Discussion

### 4.1. Functional Traits of Amblystogenium pacificum: Influence of Altitude, Sex, and Morphotype

Smaller individuals of *A. pacificum* occurred at lower altitudes, larger ones occurred at 600 m asl, and individuals with intermediate body sizes were found at 900 m asl. Since body size and body mass are often correlated in carabids [61], we were expecting a quadratic relationship between altitude and body mass. However, the observed trend was not significant. Until 600 m asl, our results were congruent with Bergmann’s law, which depicts the tendency for thermoregulation to be optimized by reaching larger adult body sizes in colder habitats, and were consistent with former studies conducted on the species [43]. At 110 m asl, which is also the altitude from which the insect start to compete with a phylogenetically related species with a smaller body size but similar diet, *Amblystogenium minimum* Luff, Davies [42,43] reported that individuals of *A. pacificum* displayed larger body sizes compared with their relatives collected at 380 m asl at Mont Branca. This author did not collect specimens at higher altitudes, making it impossible to know if a decline in the body sizes of *A. pacificum* at the 900 m site already existed in the 1980s. At 900 m asl, resource availability and temperature are substantially harsher than at lower altitudes [43], which likely explains the overall bell-shaped relationship between body size and altitude. Interestingly, the effect of altitude on body size was stronger for dark morphotypes of *A. pacificum.* This finding is consistent with the idea that higher melanism provides more light and heat absorbance, thereby participating in increasing organisms’ heating rates, thus allowing darker individuals to exhibit higher body fitness and growth rates in cold environments. Following Rensch’s rule and the study of Davies [43], we were also expecting that altitude would affect males and females differently, as the rule posits that sexual size dimorphism (SSD) decreases with size when females are larger [31]. Nevertheless, no significant differences in body size or body mass were observed between sexes or across altitude, suggesting that, in our case, Rensch’s rule did not apply. 

Intraspecific trait variations among sexes were measured for protein, lipid, and sugar amounts: females tended to have larger protein and sugar amounts than males. This difference may be related to the differential energetic needs between sexes, with females allocating more resources towards the production of reproductive reserves and tissues, including proteins and long glycogen chains [62,63,64,65]. Surprisingly, we found no differences in lipid amounts between the sexes, even though female insects usually have more lipids than males to support oogenesis [33]. In addition, neither altitude nor morphotype affected the body reserves of males and females. These results also contrast with those of Davies et al. [40], who found that dark females may reproduce more often than light ones when the temperatures are cooler, thus suggesting a greater storage of lipids for oogenesis in carabids thriving at higher altitudes. We did, however, find higher average protein amounts in light morphotypes, regardless of sex and altitude. Our expectations associated with Bergmann’s and Gloger’s rules were that we would find larger specimens with more body reserves and darker colorations at higher sites. Bergmann’s rule was met only in terms of body size, and finding higher protein contents in light morphotypes challenged our expectations for Gloger’s rule. 

### 4.2. Functional Diversity and Niche Partitioning of Amblystogenium pacificum

#### 4.2.1. Intraspecific Niche Variations and Elevation Range: Implications for Resilience

Larger hypervolume sizes are associated with broader trait ranges and higher functional richness (i.e., a higher niche space occupied by the species within a community), which in turn can suggest higher resilience of the local population of that altitude to environmental stress [8,14,66]. Here, the niche spaces occupied by beetles from most of the sampled sites had similar volumes, except for 11 m and 300 m asl, which differed from one another due to the high variation in the lipids/mass ratio at 11 m and the low proteins/mass ratio at 300 m. Apart from these two exceptions, the average variation in trait values was very similar within each site, indicating the absence of an overall pressure based on altitude, which would allow niche expansion or constraints for all traits simultaneously. 

Even if functional hypervolume sizes were similar across sites, insects from the lowest and highest altitudes occupied different functional areas, indicating that adult *A. pacificum* used the functional niche space differently at sites closer to sea level and at higher elevations. This finding suggests the occurrence of functional niche partitioning along the altitudinal gradient across dark, light, male, and female individuals. The distinct functional area occupation was mainly driven by the body size index and to a lesser extent by body mass and proteins/mass traits, which contributed less than the confidence interval expectations at each site. When put into perspective with the positive correlation between body size index and altitude, this result suggests that the sizes of *A. pacificum* are strongly selected at each sampled site, supporting the idea that altitude-driven niche constriction could be affecting the insects’ body sizes. 

In the context of global change, morphotypes with distinct functional strategies, aligned to newly encountered conditions, may become dominant [67,68,69]. In this study, we hypothesized the existence of biochemical and ecological differences between the dark and light morphotypes of *A. pacificum*, since melanism allows for better thermal absorption [70], which in turn can lead to niche partitioning. We found evidence of niche partitioning between the light and dark morphotypes: although they occupied similar parts of the functional niche space, the size of the functional niche was, on average, smaller for dark morphotypes than for light ones. For each studied altitudinal site, a part of the functional niche was unused by dark adults of *A. pacificum*. Dark insects had the smallest functional niches at higher altitudes, while lighter insects had more constrained functional niches at low altitudes. This indicated that dark specimens were functionally more specialized than their lighter counterparts, especially at higher altitudes. 

Phenotypic plasticity allows the existence of multiple coexisting intraspecific functional strategies in order to cope with environmental changes and to prevent population extinction [68]. Intraspecific trait variations can also increase population stability through the portfolio effect [71] or through reduced intraspecific competition [72,73]. In this respect, lighter morphotypes, which were more abundant at lower altitudes, displayed higher functional richness, on average, which usually confers more resilience [8,13,14,66] because species or individuals have access to a larger range of trait values to overcome disturbances or stress. Conversely, the traits for dark insects were more constrained at higher altitudes, which could confer higher vulnerability when facing competition. Therefore, it seems more likely that in this case the costs of a functional loss are in fact bypassed by the benefits of higher thermal absorbance. Moreover, the harsher environmental conditions encountered at higher altitudes may have resulted in the filtering of a restrained range of trait values that lower the functional trait variability but increase the functional specialization of insects at higher altitudes.

While males and females shared similar parts of the occupied functional niche, males had a slightly higher functional niche size when they were pooled across all sites. At each site, males were characterized by higher variability in their trait values, even though the hypervolume sizes for males were more homogeneous than those of females when examined across sites. Most traits of females collected at 900 m asl were highly constrained, causing the hypervolume size to decrease at the highest altitude for females, in contrast to the stationary hypervolume sizes for males. The decreasing functional richness for females could suggest that their traits were more constrained due to resource shortage at that altitude than in males. Body size trait constriction was apparent in males. However, the mixed models did not show a significant interaction between sex and altitude. Further replicates, especially during other seasons, would probably help ensure that the full morphological variation is detected within the two morphotypes and sexes, as the reproduction of *A. pacificum* occurs throughout the year [40].

#### 4.2.2. Intraspecific Trait Variation: True Polymorphism or Phenotypic Expression?

Despite the existence of two morphotypes of adult *A. pacificum* based on coloration, the changes in functional niche space occupation within each morphotype along the altitudinal gradient did not hold when averaging across the two morphotypes. Instead, we found that body size was more affected by the altitudinal gradient than the morphotypes were, even if the body size values of the dark morphotypes were more constrained, which reduced the proportion of the functional space they could use. 

While we did not find clear support for niche partitioning between the light and dark morphotypes, it is possible that the reduced sample sizes imposed by working with populations of an endemic species, which additionally required sampling permits, may have partly masked clear morphological patterns along the studied altitudinal gradient. Selecting relevant traits to measure adaptive adjustments is also not an easy task, especially when effects can also be due to other physiological, chemical, or potential phylogenetic signals that may exist between morphotypes. Here, it is unlikely that the sole process in action would consist of the simple phenotypic expression of organisms in response to different environmental factors, as in the heterophylla of plants or the different colorations of domestic cats [5,74,75,76,77]. Our results thus suggest that the populations of *A. pacificum* sampled along the altitudinal gradient have been subjected to selection for given trait values, whereby having an increased body size and higher melanism at elevated sites seems to provide a phenotypic advantage. The darker coloration may have resulted from functional specialization, rather than resulting from a strategy that simultaneously affects the other traits. 

The reasons for the existence of the light and dark morphotypes of *A. pacificum* are likely multiple. A common explanation is that alternative morphotypes can confer important selective advantages in different habitats. For example, Morissette et al. [78] described two different morphotypes of the lake trout *Salvinus namaycush* Walbaum (1792) based on their piscivorous versus planktivorous diets. In our case, the only habitat differences among the sites were due to altitude, and our results suggest that the light and dark morphotypes of *A. pacificum* may result from functional specialization due to the harsh environmental conditions encountered at higher altitudes, rather than from intra- or interspecific competition. 

In this regard, Sawanson et al. [72] suggested that niche partitioning facilitates coexistence and promotes higher functional diversity at the species or community scales. The larger hypervolume sizes we found when all morphotypes were pooled are consistent with this idea and revealed that they were associated with a loss of functional richness for a given morphotype (or species). Hence, the feature that contributed most to the functional niche size for all morphotypes and sexes was the body size index. Biochemical traits contributed to the niche volume, independently of the altitudinal gradient, and more in relation to the sex of the adult specimens of *A. pacificum*.

Clarifying whether the two morphotypes of *A. pacificum* are genotypically adapted to specific environmental conditions or result from phenotypic plasticity or sexual selection would require further analyses that are beyond the scope of this study. A promising perspective could be to investigate whether the morphotypes of *A. pacificum* have different behaviors in terms of resource foraging and how this might be affected by their microhabitats. Indeed, with the resource shortage being particularly strong at the higher sites of our altitudinal gradient, it is likely to be an important driver of niche segregation. Resource shortages are likely to impact the feeding behaviors of organisms that manage to survive in such harsh conditions. Feeding and foraging behaviors have been shown to lead to character displacement and to be an early stage in the development of polymorphisms [79,80].

## 5. Conclusion

In this study, we tested whether morphological and biochemical traits were related to morphotype, altitude, and sex in sub-Antarctic adults of *A. pacificum* and tested for functional niche partitioning between morphotypes, sexes, and altitudes. 

We found that all insect body sizes were positively correlated with the altitudinal gradient, and females had higher protein and sugar amounts than males. Functional hypervolume analyses did not show clear niche partitioning of the insects along the studied altitudinal gradient, and body size was the main driver of intraspecific segregation. However, darker morphotypes tended to be smaller and more functionally constrained at higher altitudes, and females showed limited trait variation at the highest altitude. Hence, we concluded that niche partitioning was happening for body sizes rather than morphotypes.

Overall, the absence of generic patterns in the temperature–size rule underlines the complex interactions of insects with the environmental conditions in their habitats, as environmental constraints (low temperatures) and distributions of trophic reserves (the low availability of resources at higher altitudes) can surely have synergistic effects on functional traits. Future efforts involving, for example, behavioral or genetical traits, and aiming to measure the reproductive success of *A. pacificum* in relation to the two morphotypes could help in understanding their occurrence along the altitudinal gradient. Such investigations are a prerequisite if we are aiming to use this species as a model case for studies investigating population range shifts in relation to global changes.

## Figures and Tables

**Figure 1 insects-14-00123-f001:**
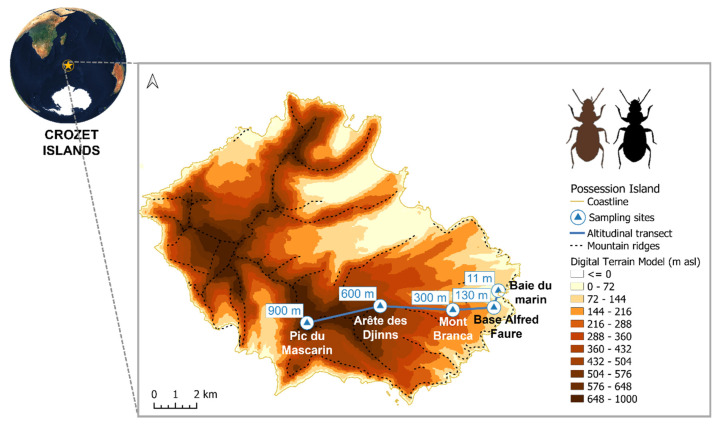
Location of Crozet Islands and description of the altitudinal transect (blue line) conducted on Possession Island (Crozet Islands). The altitudinal transect was set up from 11 m to 900 m above sea level (asl). It included five collection sites: (1) Baie du Marin at 11 m, (2) Base Alfred Faure at 130 m, (3) Mont Branca at 300 m, (4) Arête des Djinns at 600 m, and (5) Pic du Mascarin at 900 m (5). The digital elevation model was derived from a Shuttle Radar Topography Mission satellite image with a spatial resolution of 30 m, and allowed obtaining a digital terrain model.

**Figure 2 insects-14-00123-f002:**
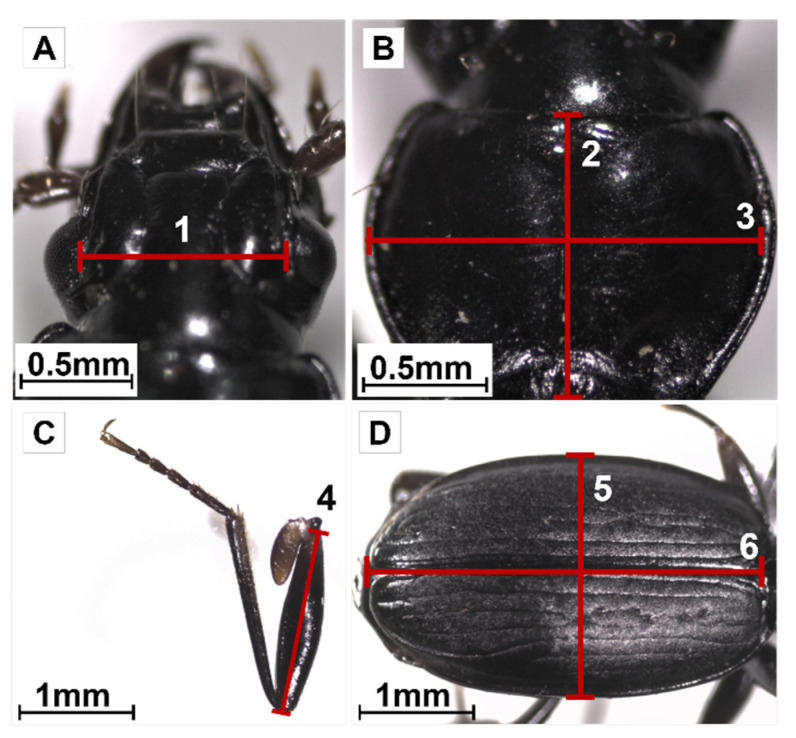
Morphometric measurements performed on adults of *Amblystogenium pacificum*. The illustrations were made on a male from the dark morphotype collected at 900 m asl. (**A**)—Head: interocular distance (1); (**B**)—Pronotum: length (2) and width (3); (**C**)—Left leg of the third pair of legs: length of the femur (4); (**D**)—Elytra: width (5) and length (6).

**Figure 3 insects-14-00123-f003:**
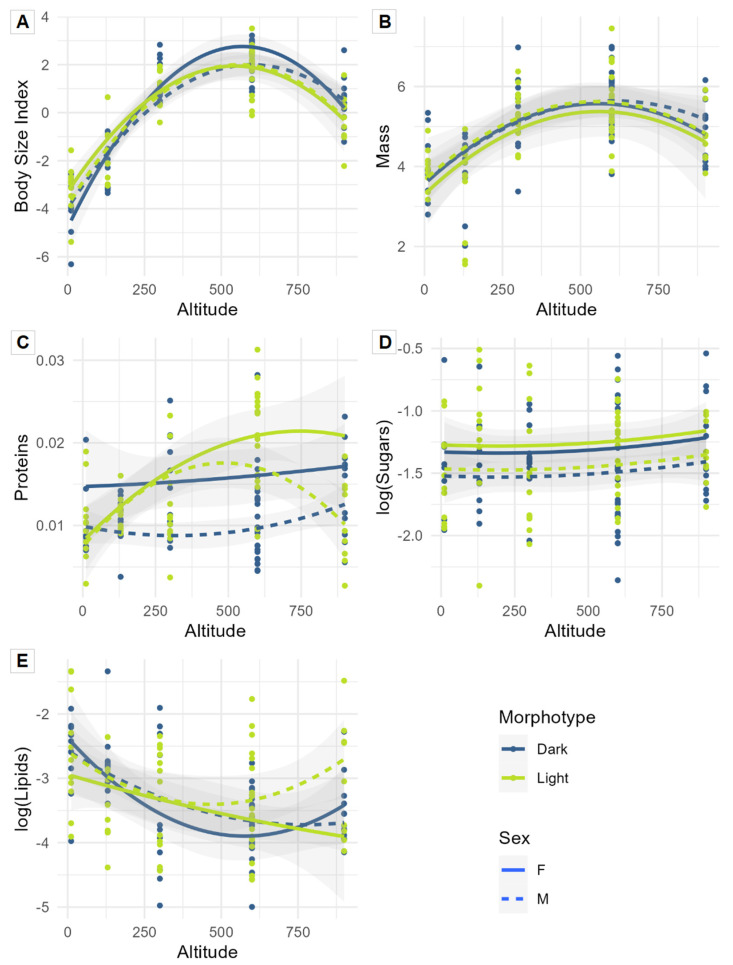
Predicted relationships (linear mixed models with site as a random factor) between altitude and the five measured functional traits for each sex (females: solid line, males: dashed line) and morphotype (dark morphotype: blue line; light morphotype: green line). Response variables: (**A**)—Body size index; (**B**)—Mass; (**C**)—Proteins/mass ratio; (**D**)—Sugars/mass ratio; and (**E**)—Lipids/mass ratio. The two last ratios were log-transformed.

**Figure 4 insects-14-00123-f004:**
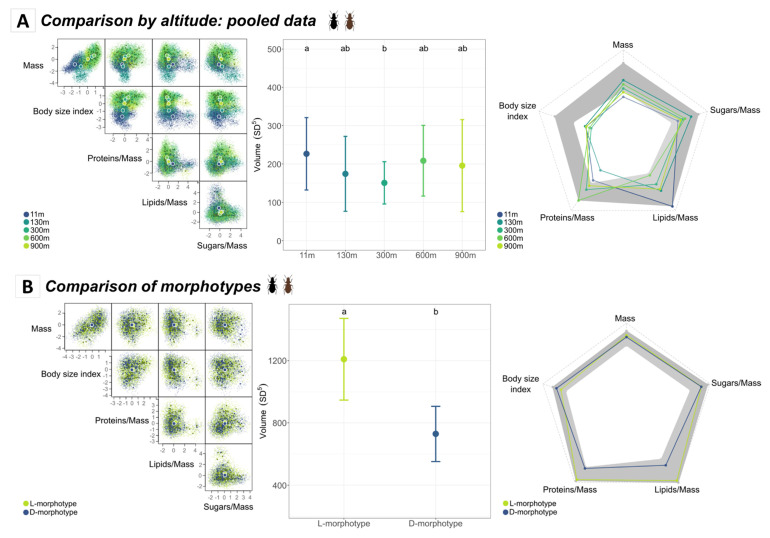
Hypervolumes based on five functional traits over five altitude levels at the species level: (**A**)—Effect of altitude on the functional hypervolume of *A. pacificum*; (**B**)—Effect of morphotype (L: light; D: dark) on the functional hypervolume; (**C**)—Effect of sex. The figures in the left panel represent the probabilistic hypervolume clouds. Colors correspond to the levels of the explanatory variables (e.g., altitude, morphotype, and/or sex), and larger points are the centroids of each cloud relative to its considered level. The different windows show the hypervolumes on a two-dimensional scale based on the corresponding trait axes mentioned beneath. The middle panel represents the *Z*-test results for the sizes of the hypervolume comparisons across the different levels of the explanatory variables. Error bars correspond to the standard deviation calculated from 100 bootstrapped hypervolumes with values resampled within each level. Letters indicate significant differences (Bonferroni-adjusted *p*-value < 0.05). The right panel shows the contributions of each trait to the sizes of the hypervolumes, such that each color is mapped to one level of the variable. The gray area corresponds to the confidence interval calculated from the bootstrapped hypervolumes for each trait.

**Figure 5 insects-14-00123-f005:**
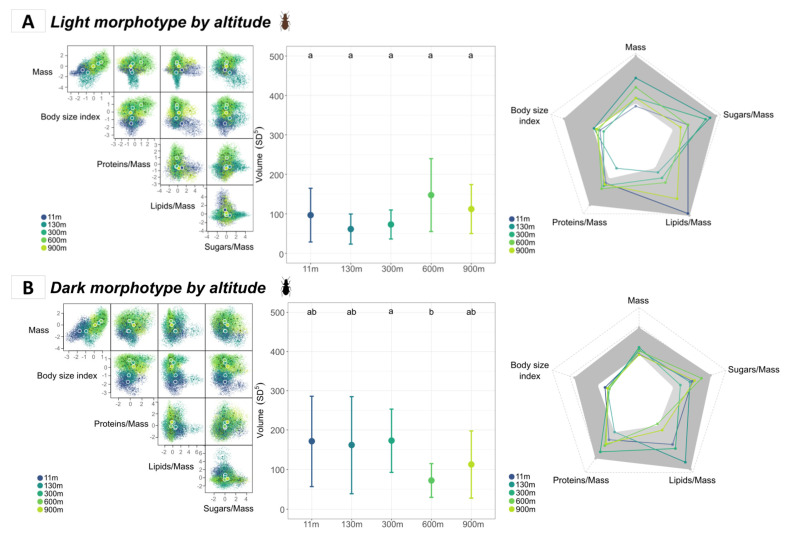
Hypervolumes based on five functional traits over five altitude levels at the species level: (**A**)—Effect of altitude on the light morphotype; (**B**)—Effect of altitude on the dark morphotype; (**C**)—Effect of altitude for females; (**D**)—Effect of altitude for males. The figures in the left panel represent the probabilistic hypervolume clouds. Colors correspond to levels of the explanatory variables (e.g., altitude, morphotype, and/or sex), and larger points are the centroids of each cloud relative to its considered level. The different cells show the hypervolumes on a two-dimensional scale based on the corresponding trait axes mentioned beneath. The middle panel represents the Z-test results on hypervolume sizes across the different levels of explanatory variables. Error bars correspond to the standard deviation calculated from 100 bootstrapped hypervolumes with values resampled within each level. Letters indicate significant differences (Bonferroni-adjusted *p*-value < 0.05). The right panel shows the contributions of each trait to the sizes of the hypervolumes at each altitude level. Colors are mapped to altitude levels; the gray area corresponds to the confidence interval calculated from the bootstrapped hypervolumes for each trait.

**Table 1 insects-14-00123-t001:** Effects of altitude, morphotype, sex, and their interactions on each functional trait. This table presents the results from the full averages of the best-fitting linear mixed-effects models selected through an AIC comparison. Functional traits are used as response variables in the following tables: **A**—Body size index; **B**—Mass; **C**—Proteins/mass; **D**—Sugars/mass; and **E**—Lipids/mass. A bold font indicates statistical significance. Std. Error—standard error; Adjusted SE—adjusted standard error; *Z* value—test statistic; Pr(>|z|)—*p*-value from the *Z* test. The colon “:” indicates the interactions among the considered variables, while “^2^” indicates a quadratic effect.

	Estimate	Std. Error	Adjusted SE	*Z* value	Pr(>|z|)
Body size index					
(Intercept)	1.881	0.262	0.264	7.130	<0.001
**Altitude**	**1.759**	**0.159**	**0.348**	**5.050**	**<0.001**
**Altitude ^2^**	**−1.712**	**0.167**	**0.367**	**4.660**	**<0.001**
L-morphotype	−0.079	0.163	0.164	0.480	0.629
Males	−0.071	0.138	0.139	0.510	0.610
**L-morphotype:Altitude**	**−0.399**	**0.165**	**0.166**	**2.400**	**0.016**
B. Mass					
(Intercept)	5.373	0.279	0.281	19.090	<0.001
Altitude	0.541	0.152	0.335	1.620	0.110
Altitude ^2^	−0.550	0.184	0.404	1.360	0.170
L-morphotype	−0.043	0.103	0.104	0.410	0.680
Males	0.083	0.136	0.137	0.610	0.540
C. Proteins/Mass					
(Intercept)	0.016	1.109 × 10^−3^	1.119 × 10^−3^	14.450	<0.001
Altitude	1.557 × 10^−3^	1.089 × 10^−3^	2.973 × 10^−3^	0.830	0.406
Altitude ^2^	−5.380 × 10^−4^	7.240 × 10^−4^	1.562 × 10^−3^	0.340	0.731
**L-morphotype**	**5.334 × 10^−3^**	**1.718 × 10^−3^**	**1.731 × 10^−3^**	**3.080**	**2.100 × 10^−3^**
**Males**	**−5.054 × 10^−3^**	**1.329 × 10^−3^**	**1.338 × 10^−3^**	**3.780**	**2.000 × 10^−4^**
Altitude:L-morphotype	1.470 × 10^−3^	1.100 × 10^−3^	1.100 × 10^−3^	1.330	0.183
Altitude:Males	−9.330 × 10^−4^	1.070 × 10^−3^	1.070 × 10^−3^	0.870	0.384
L-morphotype:Males	1.600 × 10^−3^	2.010 × 10^−3^	2.020 × 10^−3^	0.790	0.428
L-morphotype:Altitude ^2^	−3.390 × 10^−3^	1.780 × 10^−3^	1.78 0× 10^−3^	0.870	0.057
D. Sugars/Mass					
(Intercept)	−1.291	0.059	0.060	21.650	<0.001
Altitude	9.930 × 10^−3^	0.024	0.033	0.300	0.765
Altitude ^2^	3.410 × 10^−3^	0.018	0.027	0.130	0.900
L-morphotype	9.320 × 10^−3^	0.036	0.036	0.260	0.795
**Males**	**−0.190**	**0.070**	**0.071**	**2.680**	**7.400 × 10^−3^**
E. Lipids/Mass					
(Intercept)	−3.689	0.140	0.141	26.130	<0.001
Altitude	−0.354	0.095	0.183	1.930	0.054
Altitude ^2^	0.259	0.074	0.161	1.610	0.107
L-morphotype	0.132	0.149	0.150	0.880	0.377
Males	0.133	0.153	0.153	0.870	0.386
Altitude:L-morphotype	0.180	0.157	0.158	1.140	0.256
L-morphotype:Males	0.035	0.130	0.131	0.270	0.790

## Data Availability

The data presented in this study are openly available on FigShare at https://doi.org/10.6084/m9.figshare.21444753.v1 and on GitHub at https://github.com/CamilleCoux/A_pacificum/tree/main/data. The R scripts from the analyses in this article are available on GitHub at https://github.com/CamilleCoux/A_pacificum.

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
