# Peer review of "Functional Niche Partitioning Occurs over Body Size but Not Nutrient Reserves nor Melanism in a Polar Carabid Beetle along an Altitudinal Gradient"

_insects, 2023, doi:10.3390/insects14020123_

Round 1
Reviewer 1 Report
In this paper, carabid beetles were collected from 5 sites with different altitude, and their functional traits were compared among sites. From the results, authors claimed that the niche partitioning along altitudinal gradient was mainly caused by body size in this beetle species. Three statistical methods were used to test authors' hypotheses, and authors should reconsider the necessity and validity of each method.
1. FAMD
FAMD was used to explore the relationship among measured traits. FAMD can't detect a non-linear relationship between variables, and authors described the simple relationship between variables as L323-327. However, authors showed non-linear relationship between measured traits and altitude afterwards in LME model analyses. This is confusing. In addition, authors described that "most of the females may store more sugars and proteins than males, with regard to their location on the positive part of the second axis (L336-337)". This description is weak because FAMD can't show the clear relationship among variables. Difference in some traits between sexes, for example, can be shown by univariate approach, and authors did it by LME models. Results of LME models show more complex interactions among variables, therefore, simple trends described here seems not necessary for discussion. Actually, results of FAMD were not mentioned in discussion part of the paper.
2. Interaction term in LME models
Table 1 show that the interaction term was included in some models. It looks like the interaction was assumed between the linear term of altitude and other variables. Why quadratic term of altitude was not assumed to interact with other variables?
3. Presentation of predictions
Results described in Table 1B is rather simple; non-linear effect of altitude and difference between 2 morphotypes and between sexes. These descriptions correspond to results described in Fig. 4B. Results described in Table 1D is similar, however, results described in Fig. 4D is complex and it does not correspond to Table 1D. Authors should check predicted values carefully.
4. Hypervolume analyses
Results of LME model analyses showed complex interactions among trait variables. Is it proper to test the effect of each variable one by one in hypervolume analyses as described in L303-308 and as shown in Figs. 5 & 6? Authors should show the validity of this way of multiple application of this method.
Minor comments
L249 'LME' R package
Name of the package is "lme4".
L301 "contrition" -> "constriction"?
L534 "were more homogeneous that those of" -> "were more homogeneous than those of"?
L539 "contrition" -> "constriction"?
Author Response
Comments and Suggestions for Authors
In this paper, carabid beetles were collected from 5 sites with different altitude, and their functional traits were compared among sites. From the results, authors claimed that the niche partitioning along altitudinal gradient was mainly caused by body size in this beetle species. Three statistical methods were used to test authors' hypotheses, and authors should reconsider the necessity and validity of each method.
Reply: Thank you for your time and comments, we have read them carefully and believe we addressed them sufficiently. To answer your concerns about our statistical methods, we moved the FAMD analysis to the Appendix, re-ran the LME analyses and modified one of the figures (see below for more details). We thank you for your careful reading, which identified some mistakes we overlooked!
- FAMD
FAMD was used to explore the relationship among measured traits. FAMD can't detect a non-linear relationship between variables, and authors described the simple relationship between variables as L323-327. However, authors showed non-linear relationship between measured traits and altitude afterwards in LME model analyses. This is confusing. In addition, authors described that "most of the females may store more sugars and proteins than males, with regard to their location on the positive part of the second axis (L336-337)". This description is weak because FAMD can't show the clear relationship among variables. Difference in some traits between sexes, for example, can be shown by univariate approach, and authors did it by LME models. Results of LME models show more complex interactions among variables, therefore, simple trends described here seems not necessary for discussion. Actually, results of FAMD were not mentioned in discussion part of the paper.
Reply: Thank you for your comments. We agree with your interpretation here. Our original ambition was to use the FAMD as an exploratory analysis, which we would then expand upon further by the LME univariate models, and then further still with the hypervolume approach. We understand how this could create confusion for a reader, while not adding much novelty in the results and therefore now transferred the FAMD analysis to an Appendix. In this Appendix, we also included the calculation of the Body Size index based on the PCA analysis.
- Interaction term in LME models
Table 1 shows that the interaction term was included in some models. It looks like the interaction was assumed between the linear term of altitude and other variables. Why quadratic term of altitude was not assumed to interact with other variables?
Reply: This is a very good question; the exclusion of the interaction between the quadratic term of altitude and the other variables was an oversight on our end. We re-ran the linear mixed effect models after including the two-way interactions between Altitude2, Morphotype and Sex in each model. After model selection and averaging steps, all outputs remained identical, except for the model using Protein/Mass as a response variable. We included these changes in Table 1, in the Results section (L 335-336) and in the Discussion (L364-370). We are very grateful to the reviewer for having picked this up!
- Presentation of predictions
Results described in Table 1B is rather simple; non-linear effect of altitude and difference between 2 morphotypes and between sexes. These descriptions correspond to results described in Fig. 4B. Results described in Table 1D is similar, however, results described in Fig. 4D is complex and it does not correspond to Table 1D. Authors should check predicted values carefully.
Reply: Again, this is true and allowed us to identify a flaw in our plotting method: in the previous version, interaction effects were shown, even when they had not been retained after the model selection and averaging steps. Thank you very much for noticing this. We have now re-plotted the results from Table 1D such that Figure 4D shows the effects of altitude, sex and morphotype on the (log-)sugar contents of the insects without interaction effects: slopes are now identical across levels of sex and morphotype.
- Hypervolume analyses
Results of LME model analyses showed complex interactions among trait variables. Is it proper to test the effect of each variable one by one in hypervolume analyses as described in L303-308 and as shown in Figs. 5 & 6? Authors should show the validity of this way of multiple application of this method.
Reply: This was our way of testing for interaction effects, i.e., the combined effects of sex and morphotype in each site. In a regression context, it is indeed preferred to test for interactions in a single model rather than by running the model several times on subsets of the data, because the latter can increase the risk of Type I error, i.e., finding false positives. In the context of hypervolumes, however, testing for interactions in a single model is not possible. To minimize the risk for Type I errors, we used a Bonferroni correction. We hope that these precautions are enough, but we would happily accept any suggestions to further minimize this risk.
Minor comments
L249 'LME' R package
Name of the package is "lme4".
L301 "contrition" -> "constriction"?
L534 "were more homogeneous that those of" -> "were more homogeneous than those of"?
L539 "contrition" -> "constriction"?
Reply: Thank you, we have made the suggested minor corrections directly in the main text.

Reviewer 2 Report
Review of Espel et al. for Insects
Overview
This is an interesting paper in which Amblyostogenium pacificum beetles were sampled from an altitudinal transect in the sub-Antarctic Crozet islands and subject to extensive niche analysis. Analyses included morphometric data, biochemical data (% proteins, sugars and lipids), colour morph data and beetle sex. The data generally supported Bergmann’s rule showing a quadratic relationship between beetle size and altitude. Rensch’s rule did not apply. Females tended to have different chemical compositions to males, but unusually no differences in lipids were observed. Overall niche differentiation between colour morphs did not occur, but there was constrained niche-space for darker morphs at altitude. Overall, this MS is well written, the experiments well performed, the univariate and multivariate analyses seem appropriate and well-executed. The figures are excellent. Subject to some minor corrections – see below – I recommend that this MS be published in Insects.
Minor critiques
Line 33-34 “This intraspecific………” This sounds dangerously like group selectionism. Please re-write.
Line 36-37 “specimen” should read “specimens”
Line 47 “Carabidae” is a family and, therefore, should not be italicised.
Line 94 “decreased” should read “decreases”
Line 97-98 This sentence is unclear. Please re-write it.
Line 105 Check with journal style guide – do you require a taxonomic authority and year of first description on first mention of the binomial? If so, please add here. Trechidae should not be italicised as it is not a binomial. Is Trechidae even correct? I thought the Trechini was a tribe not a family, so the ending should be “i” not “idae”????
Figure 1 In Figure 1 you mention a “digital terrain model” suggest that you change this to “digital elevation model”
Line 183 Could this have been biased due to better camouflage of one morph or the other. Could size have been biased in samples due to conspicuousness of larger individuals. All these issues need to be addressed.
Line 263 Spell out generic in heading and italicise.
Line 284 Why log-transformed? – briefly explain the reason for this here.
Line 338 Is this significant? There seems to be a lot of overlap between colour morphs in Fig 3C.
Table 1 Dot notation to indicate multiplied by is not good when you have decimal points – too confusing, please re-write either with “x” or “*”.
Line 367 Spell out generic in full in a heading and also italicise the binomial.
Figure 5 A heading instead of “per altitude” say “by altitude”.
Figure 6 A B C and D change headings from “per altitude” to “by altitude”.
Line 446 Spell out generic in heading. Use normal type for binomial to differentiate it from the italic heading.
Line 453 Cite previous studies here.
Line 482 Spell out generic and use normal typescript.
Line 569 Insert taxonomic authority and year of description for this binomial.
Line 609 “reproduction” should read “reproductive”.
Author Response
Overview
This is an interesting paper in which Amblyostogenium pacificum beetles were sampled from an altitudinal transect in the sub-Antarctic Crozet islands and subject to extensive niche analysis. Analyses included morphometric data, biochemical data (% proteins, sugars and lipids), colour morph data and beetle sex. The data generally supported Bergmann’s rule showing a quadratic relationship between beetle size and altitude. Rensch’s rule did not apply. Females tended to have different chemical compositions to males, but unusually no differences in lipids were observed. Overall niche differentiation between colour morphs did not occur, but there was constrained niche-space for darker morphs at altitude. Overall, this MS is well written, the experiments well performed, the univariate and multivariate analyses seem appropriate and well-executed. The figures are excellent. Subject to some minor corrections – see below – I recommend that this MS be published in Insects.
Reply: Thank you for your support and suggestions. We addressed most of your comments directly in the main text, and believe that it has now gained in clarity and is more conform to taxonomic conventions.
Minor critiques
Line 33-34 “This intraspecific………” This sounds dangerously like group selectionism. Please re-write.
Reply: Indeed. We rephrased to “The existence of intraspecific partitioning of functional niches”.
Line 36-37 “specimen” should read “specimens”
Reply: done
Line 47 “Carabidae” is a family and, therefore, should not be italicised.
Reply: done
Line 94 “decreased” should read “decreases”
Reply: done
Line 97-98 This sentence is unclear. Please re-write it.
Reply: We have reworded this sentence and hope that it is now clearer.
Line 105 Check with journal style guide – do you require a taxonomic authority and year of first description on first mention of the binomial? If so, please add here. Trechidae should not be italicised as it is not a binomial. Is Trechidae even correct? I thought the Trechini was a tribe not a family, so the ending should be “i” not “idae”????
Reply: You are right, we added the necessary information and corrected the typos.
Changed now to: the carabid beetle Amblystogenium pacificum Putzeys 1869 (Carabidae) L105
Figure 1 In Figure 1 you mention a “digital terrain model” suggest that you change this to “digital elevation model”
Reply: done
Line 183 Could this have been biased due to better camouflage of one morph or the other. Could size have been biased in samples due to conspicuousness of larger individuals. All these issues need to be addressed.
Reply: These insects most often thrive beneath stones, where they are very easy to locate and identify. They are rather large (5-7 mm long), slow moving insects and it is thus very unlikely that sampling biases may have occurred due to camouflage. We have amended the specific sentence as follows:
Adult specimens of both morphotypes can be easily observed and captured manually in the inter- and sub-stone spaces at Crozet Islands. For the study, insects were hand-collected at five sites along an elevation gradient… (L171, 173).
Line 263 Spell out generic in heading and italicise.
Reply: done
Line 284 Why log-transformed? – briefly explain the reason for this here.
Reply: We have added a short explanation: log-transformation allowed the Sugars/Mass and Lipids/Mass variables to follow a normal distribution, which is a pre-requisite to run regressions. (L255)
Line 338 Is this significant? There seems to be a lot of overlap between colour morphs in Fig 3C.
Reply: The FAMD analysis and results are now in Appendix 2. As stated in response to Reviewer 1 as well, this analysis was more exploratory and the significance of the results does not have a precise threshold like p-values, such that significance should be evaluated visually. We agree that in the case of the morphotypes, there is indeed a lot of overlap in Fig 3C (now A2.C), such that a strong claim cannot be made.
Table 1 Dot notation to indicate multiplied by is not good when you have decimal points – too confusing, please re-write either with “x” or “*”.
Reply: All dots have been replaced by “*”.
Line 367 Spell out generic in full in a heading and also italicise the binomial.
Figure 5 A heading instead of “per altitude” say “by altitude”.
Figure 6 A B C and D change headings from “per altitude” to “by altitude”.
Line 446 Spell out generic in heading. Use normal type for binomial to differentiate it from the italic heading.
Line 453 Cite previous studies here.
Line 482 Spell out generic and use normal typescript.
Line 569 Insert taxonomic authority and year of description for this binomial.
Line 609 “reproduction” should read “reproductive”.
Thank you for your corrections. We have fixed them in the main text, highlighted by track changes.
Reviewer 2
Haut du formulaire
Overview
This is an interesting paper in which Amblyostogenium pacificum beetles were sampled from an altitudinal transect in the sub-Antarctic Crozet islands and subject to extensive niche analysis. Analyses included morphometric data, biochemical data (% proteins, sugars and lipids), colour morph data and beetle sex. The data generally supported Bergmann’s rule showing a quadratic relationship between beetle size and altitude. Rensch’s rule did not apply. Females tended to have different chemical compositions to males, but unusually no differences in lipids were observed. Overall niche differentiation between colour morphs did not occur, but there was constrained niche-space for darker morphs at altitude. Overall, this MS is well written, the experiments well performed, the univariate and multivariate analyses seem appropriate and well-executed. The figures are excellent. Subject to some minor corrections – see below – I recommend that this MS be published in Insects.
Reply: Thank you for your support and suggestions. We addressed most of your comments directly in the main text, and believe that it has now gained in clarity and is more conform to taxonomic conventions.
Minor critiques
Line 33-34 “This intraspecific………” This sounds dangerously like group selectionism. Please re-write.
Reply: Indeed. We rephrased to “The existence of intraspecific partitioning of functional niches”.
Line 36-37 “specimen” should read “specimens”
Reply: done
Line 47 “Carabidae” is a family and, therefore, should not be italicised.
Reply: done
Line 94 “decreased” should read “decreases”
Reply: done
Line 97-98 This sentence is unclear. Please re-write it.
Reply: We have reworded this sentence and hope that it is now clearer.
Line 105 Check with journal style guide – do you require a taxonomic authority and year of first description on first mention of the binomial? If so, please add here. Trechidae should not be italicised as it is not a binomial. Is Trechidae even correct? I thought the Trechini was a tribe not a family, so the ending should be “i” not “idae”????
Reply: You are right, we added the necessary information and corrected the typos.
Changed now to: the carabid beetle Amblystogenium pacificum Putzeys 1869 (Carabidae) L105
Figure 1 In Figure 1 you mention a “digital terrain model” suggest that you change this to “digital elevation model”
Reply: done
Line 183 Could this have been biased due to better camouflage of one morph or the other. Could size have been biased in samples due to conspicuousness of larger individuals. All these issues need to be addressed.
Reply: These insects most often thrive beneath stones, where they are very easy to locate and identify. They are rather large (5-7 mm long), slow moving insects and it is thus very unlikely that sampling biases may have occurred due to camouflage. We have amended the specific sentence as follows:
Adult specimens of both morphotypes can be easily observed and captured manually in the inter- and sub-stone spaces at Crozet Islands. For the study, insects were hand-collected at five sites along an elevation gradient… (L171, 173).
Line 263 Spell out generic in heading and italicise.
Reply: done
Line 284 Why log-transformed? – briefly explain the reason for this here.
Reply: We have added a short explanation: log-transformation allowed the Sugars/Mass and Lipids/Mass variables to follow a normal distribution, which is a pre-requisite to run regressions. (L255)
Line 338 Is this significant? There seems to be a lot of overlap between colour morphs in Fig 3C.
Reply: The FAMD analysis and results are now in Appendix 2. As stated in response to Reviewer 1 as well, this analysis was more exploratory and the significance of the results does not have a precise threshold like p-values, such that significance should be evaluated visually. We agree that in the case of the morphotypes, there is indeed a lot of overlap in Fig 3C (now A2.C), such that a strong claim cannot be made.
Table 1 Dot notation to indicate multiplied by is not good when you have decimal points – too confusing, please re-write either with “x” or “*”.
Reply: All dots have been replaced by “*”.
Line 367 Spell out generic in full in a heading and also italicise the binomial.
Figure 5 A heading instead of “per altitude” say “by altitude”.
Figure 6 A B C and D change headings from “per altitude” to “by altitude”.
Line 446 Spell out generic in heading. Use normal type for binomial to differentiate it from the italic heading.
Line 453 Cite previous studies here.
Line 482 Spell out generic and use normal typescript.
Line 569 Insert taxonomic authority and year of description for this binomial.
Line 609 “reproduction” should read “reproductive”.
Thank you for your corrections. We have fixed them in the main text, highlighted by track changes.

Round 2
Reviewer 1 Report
This paper was improved from the previous version though I still have doubts about the validity of hyper volume analyses.
My previous comments:
Results of LME model analyses showed complex interactions among trait variables. Is it proper to test the effect of each variable one by one in hypervolume analyses as described in L303-308 and as shown in Figs. 5 & 6? Authors should show the validity of this way of multiple application of this method.
Authors' response:
This was our way of testing for interaction effects, i.e., the combined effects of sex and morphotype in each site. In a regression context, it is indeed preferred to test for interactions in a single model rather than by running the model several times on subsets of the data, because the latter can increase the risk of Type I error, i.e., finding false positives. In the context of hypervolumes, however, testing for interactions in a single model is not possible. To minimize the risk for Type I errors, we used a Bonferroni correction. We hope that these precautions are enough, but we would happily accept any suggestions to further minimize this risk.
Authors described the result of analyses for pooled data in L349-367, and described the result of analyses for separated data in L379-399. Discussion was also made in this context, description of Fig. 5A in L475-476 and description of Fig. 6CD in L521-523 for example. This is confusing because the trend is different between morphotypes and sexes. Is the analysis of Fig. 5A is necessary for the objective to test "the combined effects of sex and morphotype in each site"?
In addition, the trend described in L394-396 "Conversely, males tended to have a positive quadratic response to increasing altitude, with the minimal volume corresponding to the 130 m site being significantly lower than the 900 m site hypervolume size, which was its maximal size" does not correspond to the middle panel of Fig. 6D. I'm sorry but I didn't notice this point previously.
Minor comments:
Table 1
Indent was set for the variable list of the table. I think a hanging indent is better for the discrimination of each model.
Author Response
This paper was improved from the previous version though I still have doubts about the validity of hyper volume analyses.
Reply : Thank you for helping us to improve the previous version. We regret that the Bonferroni correction was insufficient to entirely remove your doubts about the hypervolume analysis.
My previous comments:
Results of LME model analyses showed complex interactions among trait variables. Is it proper to test the effect of each variable one by one in hypervolume analyses as described in L303-308 and as shown in Figs. 5 & 6? Authors should show the validity of this way of multiple application of this method.
Authors' response:
This was our way of testing for interaction effects, i.e., the combined effects of sex and morphotype in each site. In a regression context, it is indeed preferred to test for interactions in a single model rather than by running the model several times on subsets of the data, because the latter can increase the risk of Type I error, i.e., finding false positives. In the context of hypervolumes, however, testing for interactions in a single model is not possible. To minimize the risk for Type I errors, we used a Bonferroni correction. We hope that these precautions are enough, but we would happily accept any suggestions to further minimize this risk.
Reply : Unfortunaltely, it is difficult for us to provide more arguments without further guidance about what remains unconvincing. If we misunderstood your doubts or perhaps only partially understood them, it would help to know which aspect.
Authors described the result of analyses for pooled data in L349-367, and described the result of analyses for separated data in L379-399. Discussion was also made in this context, description of Fig. 5A in L475-476 and description of Fig. 6CD in L521-523 for example. This is confusing because the trend is different between morphotypes and sexes. Is the analysis of Fig. 5A is necessary for the objective to test "the combined effects of sex and morphotype in each site"?
Reply : We understand that these results might be confusing, as they do appear contradictory. But this is precisely why we believe that it is necessary to keep both the pooled data analysis and the separated data analysis in the study: the zooming in from the larger scale of the pooled data to the finer sclae of the separated groups allowed to unveil more complex mechanisms that may cancel each other out at other scales.
In addition, the trend described in L394-396 "Conversely, males tended to have a positive quadratic response to increasing altitude, with the minimal volume corresponding to the 130 m site being significantly lower than the 900 m site hypervolume size, which was its maximal size" does not correspond to the middle panel of Fig. 6D. I'm sorry but I didn't notice this point previously.
Reply : This is true, thank you very much for noticing. We have now edited the main text as follows : « Conversely, males tended to have a positive quadratic response to increasing altitude, with the minimal volume corresponding to the 130 m site being significantly lower than the 11 m site hypervolume size, which was its maximal size, and lower than the 900 m site. » (L395-398).
Minor comments:
Table 1
Indent was set for the variable list of the table. I think a hanging indent is better for the discrimination of each model.
Reply : We have followed this suggestion nand hope that the models are now more easily distinguished.
